# Sociodemographic Factors Associated with Vaccine Hesitancy in Central Texas Immediately Prior to COVID-19 Vaccine Availability

**DOI:** 10.3390/ijerph19010368

**Published:** 2021-12-30

**Authors:** John R. Litaker, Naomi Tamez, Carlos Lopez Bray, Wesley Durkalski, Richard Taylor

**Affiliations:** 1Office of Population Health and Science, The Litaker Group, LLC, Austin, TX 78716, USA; 2Office of Population Health, Sendero Health Plans, Inc., Austin, TX 78741, USA; Naomi.Tamez@senderohealth.com (N.T.); Carlos.Bray@senderohealth.com (C.L.B.); 3Sendero Health Plans, Inc., Austin, TX 78741, USA; Wesley.Durkalski@senderohealth.com; 4Undergraduate Public Health Program, University of Texas at Austin, Austin, TX 78712, USA; RTaylor65@utexas.edu

**Keywords:** COVID-19, SARS-CoV-2, vaccine hesitancy, vaccination

## Abstract

Vaccine-induced herd immunity remains the best opportunity for ending the COVID-19 pandemic. However, COVID-19 vaccine hesitancy is a real concern. In this paper, we report on vaccine hesitancy in Central Texas immediately prior to the release of the two mRNA COVID-19 vaccines in late December 2020. A total of 1648 individuals 18 years or older with health insurance living in Central Texas completed a survey on sociodemographic factors and plans to obtain the COVID-19 vaccine. Of the respondents, 64.1% planned to obtain the COVID-19 vaccine. Logistic regression identified the following sociodemographic factors associated with vaccine hesitancy: Black or African American race (POR: 0.351, *p* < 0.001, 95% CI: 0.211, 0.584), female sex (POR: 0.650, *p* < 0.001, 95% CI: 0.518, 0.816), age of 35–49 years old (POR: 0.689, *p* = 0.004, 95% CI: 0.534, 0.890), annual household income of less than US$10,000 (POR: 0.565, *p* = 0.041, 95% CI: 0.327, 0.976), a high school education or less (POR: 0.565, *p* = 0.001, 95% CI: 0.401, 0.795), and a high school education but less than a 4-year college degree (POR: 0.572, *p* < 0.001, 95% CI: 0.442, 0.739). Real-world evidence provided by individuals on plans to get vaccinated can reveal COVID-19 vaccine hesitancy associated heterogeneity.

## 1. Introduction

Vaccine hesitancy is a complicated construct. The World Health Organization working definition on vaccine hesitancy states that “vaccine attitudes can be seen on a continuum, ranging from total acceptance to complete refusal [while] vaccine-hesitant individuals are a heterogeneous group in the middle of this continuum” [1]. Complementing this definition is research that shows vaccine hesitancy is related to safety and efficacy, preference to naturopathic medicine, perceived susceptibility of disease, and lack of knowledge about vaccines [2,3,4,5,6]. The heterogeneity of vaccine-hesitant individuals will vary by a specific vaccine, the people involved, place, and time period [7] and can manifest individually as issues of confidence, complacency, or convenience [1]. A framework released in 2021 after COVID-19 vaccines had been approved further defined those in the center of the continuum as the “moveable middle” with the recognition that they could shift to become either vaccine acceptors or vaccine refusers [8].

The COVID-19 pandemic has exacerbated the issue of vaccine hesitancy. For COVID-19, there has been some disquiet about the expedited timeline for vaccine development, a process that can normally take years to occur [9,10]. Concerns have also been expressed about the regulatory review and approval process, a process that has traditionally been perceived as robust and evidence-based [11,12]. Evidence on morbidity, mortality, and hospitalization indicate worse outcomes for individuals who have not yet received the vaccine, [13] and vaccine-induced herd immunity remains the best opportunity for ending the pandemic. Vaccine hesitancy undermines the push toward herd immunity, and this problem may become exacerbated as vaccine availability widens [14].

Vaccine hesitancy is of particular concern among communities of color. Cost, trust, and confidence are cited as factors that impact decision-making among Black or African Americans with regard to seasonal influenza vaccine [15,16]. Vaccine hesitancy among Black or African Americans for the COVID-19 vaccine has been shown to be related to mistrust of the medical establishment, concerns about the vaccine development timeline, and lack of data on vaccine side effects [17]. Among Hispanics, low uptake of the seasonal influenza vaccine is due in part to lack of awareness and perceptions of individual susceptibility to disease, [18] beliefs that may also explain vaccine hesitancy associated with the COVID-19 vaccine [19].

As the COVID-19 pandemic unfolded and as Emergency Use Authorization was about to be granted for two mRNA vaccines in the United States, we wanted to identify sociodemographic factors associated with vaccine hesitancy among a population of individuals with health insurance in Central Texas. To identify such factors, Sendero Health Plans, Inc. (Sendero), a community-based, nonprofit health maintenance organization in Austin, Texas providing health insurance coverage under the Patient Protection and Affordable Care Act (ACA), surveyed its members on questions related to COVID-19 and vaccinations. In this study, we report on individual plans to obtain the COVID-19 vaccine along with social and demographic factors associated with vaccine hesitancy.

## 2. Materials and Methods

Eligible participants for this study were Sendero head-of-household members. Head-of-household members are defined as adult members 18 years old or older who are the primary policyholder for a Sendero ACA health insurance plan. Participants completed an online survey that included sociodemographic questions and questions about their plan to obtain the COVID-19 vaccine. Individuals were invited to participate either by email or by post, depending on the communication preference previously provided by each member. All individuals had a minimum of three weeks to complete the survey. The study commenced on 11 November 2020 and ended on 22 December 2020. COVID-19 vaccine distribution did not occur in Central Texas until after this survey closed. All responses were submitted using the online Qualtrics platform (Qualtrics, Provo, UT, USA). Participation in the survey was voluntary, and those who completed the survey were sent a $25 gift card to a local grocery merchant. All communication was provided in English and Spanish. Personal health information was not collected. All data were de-identified prior to analysis. Pairwise deletion was used to address cases of missing data.

Survey questions were based on a review of the peer-reviewed literature, using similar studies on vaccine hesitancy and social determinants of health constructs. Questions were reviewed to determine applicability to our study and audience. The survey was not piloted in advance.

Variables of interest included sociodemographic factors such as age, gender, race, ethnicity, level of educational attainment, and income (see Table 1). Age in years was computed using the difference between the date of survey completion and the member’s date of birth. All statements or questions except those associated with race and ethnicity required a single response. The survey allowed individuals to self-identify with multiple race or ethnic identities to reflect the diversity of respondents’ racial and ethnic heritage. Descriptions of univariate categorical variables include count, percent for each level of the variable, and quantitative variables include frequency, percent, mean, and standard deviation.

The primary outcome variable of interest was whether a respondent planned to be vaccinated with a COVID-19 vaccine when it became available. Specifically, members were asked to respond to the following statement: “I plan to get the COVID-19 vaccine when it is available”. Response options included yes, no, not sure, and prefer not to answer.

Analyses appropriate for a cross-sectional survey design were used. Unadjusted bivariate analyses were performed to describe relationships between variables and to identify statistically significant independent variables to include in the logistic regression analysis. The chi-square test for independence (χ^2^) with corresponding degrees of freedom [χ^2^(df)] and prevalence odds ratios (POR) were used to describe relationships between categorical variables. All unadjusted bivariate analyses assume the null form of no relationship between the variables.

Original survey responses were used to restructure the new dichotomous form of the dependent variable, “I plan to get the COVID-19 vaccine when it is available” to compare the yes response to all other responses (no, not sure, and prefer not to answer). Restructured independent variables included education (≤High School, >HS < 4-year college, and ≥4-year college degree) and age by years (18–34, 35–49, 50 and older). The remaining independent variables (i.e., sex, household income, Black or African American race, and Hispanic, Latino, or Spanish ethnicity) were categorical.

Logistic regression was used to examine the relationship between the dichotomized dependent variable “I plan to get the COVID-19 vaccine when it is available” and the previously identified statistically significant independent variables identified using unadjusted bivariate analyses. All statistically significant independent variables identified in the bivariate analyses were included in the logistic regression model using single-step block entry. We chose this method rather than a forward or backward stepwise approach in order to compute the impact of all selected independent variables in the regression model.

The Type I error rate was set as α = 0.05 for all unadjusted and adjusted analyses. Appropriate chi-square values with degrees of freedom, prevalence odds ratios, *p*-values, and 95% confidence intervals are reported for unadjusted and adjusted analyses. Analyses were conducted using IBM SPSS for Macintosh, Version 27.0 (Armonk, NY, USA).

## 3. Results

### 3.1. General Findings

A total of 1660 persons completed the survey for a response rate of 19.8%. Twelve persons were removed from the analysis because they were either less than 18 years of age or we could not confirm their date of birth. In total, data from 1648 persons were included in subsequent analyses. Table 2 provides a summary of demographic data obtained from the cross-sectional survey. The mean respondent age was 46.8 years old (Range: 20–86, SD ± 12.3). Individuals identified as female comprised 54.1% of respondents, and more than half of respondents (64.7%) reported having attained at least a bachelor’s degree. The majority of respondents (67.9%) reported a household income of less than $50,000 per year with 42.8% of all respondents reporting a household income of less than $30,000 per year. The majority of respondents (81.9%) self-identified as white race and 19.7% self-identified a Hispanic, Latino, or Spanish ethnicity.

For the dependent variable, the majority of respondents (64.1%) selected yes, while 5.5% and 3.8% selected no or not sure, respectively. Over one-fourth (26.6%) of respondents selected prefer not to answer. (See Figure 1).

### 3.2. Unadjusted Bivariate

Table 3 includes the results of chi-square tests of independence for all unadjusted bivariate analyses. Three dichotomous independent variables: (1) female; (2) Black or African American race; and (3) Hispanic, Latino, or Spanish ethnicity, were all independently statistically significant. Among the categorical variables, respondents aged 35–49 years old, education level of less than a 4-year college degree, and household income of less than $76,000 a year were all independently statistically significant.

### 3.3. Logistic Regression

Table 3 includes the adjusted results for the independent variables and the dependent variable. The overall logistic regression analysis associating plans to get the COVID-19 vaccine when available achieved statistical significance (χ^2^(13) = 90.60, *p* < 0.001). Hispanic, Latino, or Spanish ethnicity did not significantly contribute to the adjusted logistic regression model (POR: 0.826, *p* = 0.186, 95% CI: 0.622, 1.096).

Respondents who identified as Black or African American were 65% less likely to plan to get the COVID-19 vaccine (POR: 0.351, *p* < 0.001, 95% CI: 0.211, 0.584) than individuals who did not identify as Black or African American. Similarly, those who identified as female were 35% less likely to plan to get the COVID-19 vaccine (POR: 0.650, *p* < 0.001, 95% CI: 0.518, 0.816) than those who identified as male. Individuals who were aged 35–49 years old, as compared to those who were 50 years old or older continued to retain significance in the logistic regression analysis (POR: 0.689, *p* = 0.004, 95% CI: 0.534, 0.890).

Respondents with up to and including a high school education and those with at least a high school education but less than a 4-year college degree were 43.5% (POR: 0.565, *p* = 0.001, 95% CI: 0.401, 0.795) and 42.8% (POR: 0.572, *p* < 0.001, 95% CI: 0.442, 0.739) less likely, respectively to indicate plans to get the COVID-19 vaccine than individuals with at least a 4-year college degree. Respondents reporting an annual household income of less than $10,000 per year were 43.5% less likely to plan to get the COVID-19 vaccine (POR: 0.565, *p* = 0.041, 95% CI: 0.327, 0.976) than individuals reporting an annual household income of more than $100,000 per year.

## 4. Discussion

Vaccine hesitancy is a complex and multifactorial construct. Disentangling this construct is necessary to identify factors associated with vaccine hesitancy, reasons why individuals exhibit vaccine hesitancy, and what, if anything, can be done to move individuals from being vaccine-hesitant to becoming vaccine acceptors. In this study, we sought to disentangle the first part of this construct by identifying factors that independently predicted planned vaccine uptake among a group of 1648 people living in Central Texas.

Overall, our study showed that 64.1% of individuals who responded to our survey planned to obtain the COVID-19 vaccine when it became available. Among those who do not plan to take the vaccine, logistic regression identified five sociodemographic factors that independently predict a lack of planned uptake: Black or African American race, female sex, age 35–49 years, low educational attainment, and a low annual household income.

The finding that individuals of Black or African American race are less likely to obtain the COVID-19 vaccine is not surprising. In fact, our finding is consistent with the literature for both the COVID-19 vaccine and for the seasonal influenza vaccine [15,16,17,20]. In a study of 948 persons from North Carolina, individuals who identified as Black were more likely to be COVID-19 vaccine-hesitant than compared to the referent group (OR: 1.69, *p* = 0.002, 95% CI: 1.16, 2.45) [20]. In a study of seasonal influenza among health care workers, non-Hispanic blacks had lower uptake of vaccination than non-Hispanic whites based on a total direct effect (PR = 0.87; 95% CI = 0.75, 0.99) [21].

Our data also indicate that women are 35% less likely than men to obtain the COVID-19 vaccine. This finding is consistent with other COVID-19 studies, including the North Carolina study. Data from that study showed a higher likelihood of hesitancy toward the COVID-19 vaccine among women (OR: 1.90, *p* < 0.0001, 95% CI: 1.36, 2.64) [20]. This finding is supported by other published studies showing vaccine hesitancy for females associated with the COVID-19 vaccine, the H1N1 pandemic influenza vaccine, and the seasonal influenza vaccine [22,23,24].

Based on our finding of vaccine hesitancy among women, we hypothesized that women may be more reluctant to obtain the COVID-19 vaccine because of safety concerns related to pregnancy, birth, and the potential negative health outcomes of infants born to vaccinated mothers. In a post-hoc analysis, we stratified females (*N* = 891) by reproductive age (18–44 years old) and non-reproductive age (≥45 years old). We did not identify a statistically significant difference in planned COVID-19 vaccine uptake between these two groups (59.2% vs. 57.0%; PR = 1.03, 95% CI = 0.97, 1.08). Further research is needed to better understand why women are more hesitant than men to obtain the COVID-19 vaccine.

Regarding ethnicity, our bivariate data identified a statistically significant association between Hispanic ethnicity and a decreased likelihood to obtain the COVID-19 vaccine (*p* = 0.001); however, statistical significance was not retained in the logistic regression model (*p* = 0.186). This finding was surprising to us. We had hypothesized that there would be an association between COVID-19 vaccine hesitancy and Hispanic ethnicity based on previous research with seasonal influenza and ethnicity [18,19]. In a recent study of racial and ethnic differences for COVID-19 vaccine hesitancy among health care workers, an association between COVID-19 vaccine hesitancy and Hispanic ethnicity was found in each of the three regression models used (each *p* < 0.001) [22]. The conflicting findings between our study and the healthcare worker study once again support the idea of heterogeneity among the vaccine-hesitant and the need to assess vaccine hesitancy across people, place, and time.

To provide additional context to our findings, we consider some additional points that may influence a person’s decision to obtain the COVID-19 vaccine. Data on COVID-19 vaccine uptake noted below is based on population-level data in Travis County, Texas from the Texas Department of State Health Services (as of 21 December 2021), not Sendero membership directly. 

Firstly, general uptake of non-COVID-19 adult vaccines may provide a point of reference for uptake of the COVID-19 vaccine. Overall, 20.2% of persons ≥ 19 years of age in 2018 had received all age-appropriate vaccines [25]. For the 2017–2018 seasonal influenza vaccine, 46.1% of persons ≥ 19 years of age were vaccinated, a rate that increased to 61.0% for persons ≥ 65 years of age. For the pneumococcal vaccine, 69.0% of persons ≥ 65 years of age were vaccinated [25]. By comparison, full-dose COVID-19 vaccine coverage ranged from 19.8% to 96.3% across all 254 Texas counties, with a statewide average of 60.5% [26]. The full-dose coverage rate for the COVID-19 vaccine in Travis County, Texas (where most Sendero members reside) was 69.0% for those ≥ 5 years of age and 87% for persons ≥ 65 years of age [26]. This finding is similar to our study finding that 64.1% of members ≥ 18 years of age planned to obtain the COVID-19 vaccine when available.

A second point for consideration is that county-level data may mask underlying racial or ethnic disparities in COVID-19 vaccine uptake. While 69.0% of the Travis County population is fully vaccinated, 3.93% of the vaccinated population is Black or African American as reported by individuals at the time of vaccination. This proportion is less than half of the 8.9% estimated US Census Bureau proportion of Black or African Americans living in Travis County [27]. This finding is consistent with our finding that Blacks or African Americans are 65% less likely to get vaccinated than persons who do not identify as Black or African American. In addition, these findings are consistent with research that indicate African Americans lack trust in government due to past medical abuse associated with the Tuskegee syphilis study that intentionally denied treatment to African Americans infected with syphilis and a general unease by African Americans due to past discrimination [16].

A final point for consideration involves the uptake of the COVID-19 vaccine among women. Our data indicated that women were 35% less likely to plan to obtain the COVID-19 vaccine than men. Full-dose COVID-19 vaccine uptake data by sex indicates that women and men in Travis County, Texas are about proportionally equal in uptake at 50.3% and 49.7%, respectively.

### 4.1. Recommendations

Health Insurance Plans have an opportunity to help members manage their health and improve health outcomes. This study identified factors associated with COVID-19 vaccine hesitancy. We now know select sociodemographic characteristics associated with hesitancy. What we do not know, however, is the “why” nor do we know what, if anything, can be done to help a person transition from the moveable middle to becoming a vaccine acceptor. Further studies should be conducted to understand why a person has not obtained the COVID-19 vaccine and what could be done to change their mind to obtain the COVID-19 vaccine. This information then can be used to develop focused outreach and education to these individuals to help address perceived or real barriers and allow them to make an informed choice about getting vaccinated.

### 4.2. Limitations

Our data is based on a cross-section of individuals who have health insurance and who decided to engage with their health insurance company on this topic. Vaccine hesitancy and health are sensitive topics; therefore, individuals who responded to this survey likely differ in some ways from individuals who did not respond. We do not have situational awareness of what these differences may be. Secondly, this data represents the 1648 individuals who responded to this survey. We did not intend, nor was this survey powered to be representative of the community at large. However, within Travis County, Texas the US Census data from 2020 estimates the population to be 49.5% female (our study: 54.1%), 79.7% white (our study: 81.9%), 8.9% Black (our study: 5.0%), and 33.6% Hispanic (our study: 19.7%). Census data also estimates 50.0% of persons in Travis County, Texas have attained a bachelor’s degree (our study: 64.7%). The median income is estimated to be $75,887; however, we do not have a comparison amount from our study as income was collected as a categorical variable. Thirdly, studies that use self-reported information are at risk of user misclassification. However, unpublished data from two additional surveys conducted by Sendero Health Plans on this topic show consistent reporting of both plans to obtain the COVID-19 vaccine and seasonal influenza data. Finally, as all participants in this survey have health insurance, we do not know if there is a difference in planned COVID-19 vaccine uptake among the uninsured or underinsured; however, we believe that since COVID-19 vaccination is free at the point of service, health insurance status would not introduce sufficient bias. If anything, individuals who are uninsured or underinsured may be more hesitant to obtain the vaccine than the population described herein, as they may be less likely to interact with the health care system at large, thus increasing the need for public health to operate mass vaccination clinics [28].

## 5. Conclusions

Vaccine hesitancy, as a heterogeneous theoretical framework, has been well described. What is particularly important about this framework is that multiple factors influence the decision-making process to accept or refuse a vaccine. As such, conducting contemporary empirical research focused on COVID-19 in various communities across the United States, and indeed across the world, is necessary to understand who is hesitant, why, and how education and outreach can be focused to overcome such hesitancy.

We are in a new space and a new time with COVID-19. The SARS-CoV-2 virus that causes COVID-19 has strained health systems worldwide, halted global commerce, and resulted in lockdowns that prevented people from leaving their homes except in limited circumstances. In addition, vaccines to prevent COVID-19 disease were approved using a rarely used Emergency Use Authorization procedure. Two of these vaccines were developed using novel mRNA technology which, despite having been used successfully in animals and having a strong safety profile [29,30], had not been used in humans until Emergency Use Authorization was granted. All of these factors contribute to what the World Health Organization calls “the constellation of factors that may influence the decision to accept some or all vaccines” [1]. Therefore, it is vitally important that we conduct empirical research to help inform our current thinking and actions related to this vaccine in the people, places, and times where the COVID-19 vaccine is to be distributed and administered. Past research, while certainly appropriate for generating hypotheses, does not negate the need to address current COVID-19 vaccine hesitancy associated heterogeneity.

## Figures and Tables

**Figure 1 ijerph-19-00368-f001:**
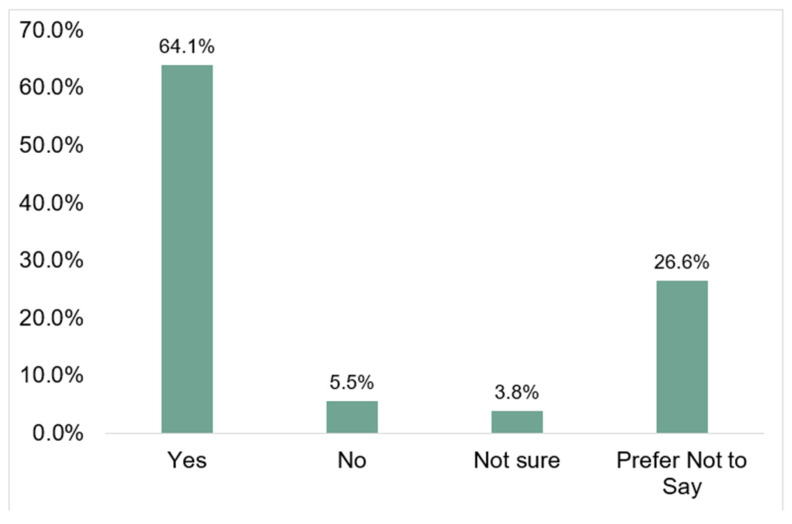
Percent of respondents who plan to obtain the COVID-19 vaccine by response category (*N* = 1648).

**Table 1 ijerph-19-00368-t001:** Reported demographic and summary characteristics of survey respondents.

Characteristics of the Respondent Population	*n* (%)
Sex (*N* = 1648)	
Female	891 (54.1)
Male	757 (45.9)
Age in Years (*N* = 1648) (Range: 20–86 years)	1648 (100)
18–24 years old	24 (1.5)
25–34 years old	327 (19.8)
35–44 years old	381 (23.1)
45–54 years old	356 (21.6)
55–64 years old	501 (30.4)
≥65 years old	59 (3.6)
Race (*N* = 1648) (More than one race response could be selected)	
American Indian or Alaskan Native	24 (1.5) *
Asian	130 (7.9) *
Black or African American	83 (5.0) *
Native Hawaiian or Other Pacific Islander	6 (0.4) *
White	1349 (81.9) *
Other	112 (6.8) *
Ethnicity (*N* = 1648)	
No, not of Hispanic, Latino, or Spanish origin	1324 (80.3)
Yes, Hispanic, Latino, or Spanish origin (More than one ethnicity could be selected)	324 (19.7)
Mexican, Mexican American, Chicano	245 (14.9) *
Puerto Rican	11 (0.7) *
Cuban	14 (0.8) *
Another Hispanic, Latino, or Spanish origin	87 (5.3) *
Education (*N* = 1648)	
Less than some high school	16 (1.0)
Some high school	55 (3.3)
High School Diploma, GED, or equivalent	162 (9.8)
Trade School	47 (2.9)
Some College	300 (18.2)
Associate Degree	132 (8.0)
Bachelor’s Degree	600 (36.4)
Graduate Degree	329 (20.0)
Other	7 (0.4)
Annual Household Income (*N* = 1648)	
Less than $10,000 per year	159 (10.4)
$10,000–$29,999	461 (28.0)
$30,000–$39,999	216 (13.1)
$40,000–$49,999	154 (9.3)
$50,000–$75,999	242 (14.7)
$76,000–$99,999	104 (6.3)
$100,000 or above	128 (7.8)
Prefer Not To Answer	172 (10.4)

* Respondents were able to represent their racial and ethnic heritage by selecting more than one racial or ethnic group. For this table, the percentages reflect a denominator of 1648 for each of these response options and may not total 100%.

**Table 2 ijerph-19-00368-t002:** Results of chi-square tests of independence of plan to receive the vaccine by selected demographic variables.

Variable	*N*	%	I Plan to Get the COVID-19 Vaccine When It Is Available?				Chi-Square (df) *p*-Value
			Yes	No	Not Sure	Prefer Not to Say	
			*n* (%)	*n* (%)	*n* (%)	*n* (%)	
Age Level (*N* = 1648)							21.52 (6), *p* = 0.001
18–34 years of age	351	21.3	226 (21.4)	13 (14.3)	13 (20.6)	99 (22.6)	
35–49 years of age	559	33.9	332 (31.4)	39 (42.9)	34 (54.0)	154 (35.2)	
50+ year of age	738	44.8	498 (47.2)	39 (42.9)	16 (25.4)	185 (42.2)	
Education Level (*N* = 1641)							30.35 (6), *p* < 0.001
≤High School Diploma ^†^	233	14.2	127 (12.0)	21 (23.1)	12 (19.4)	73 (16.8)	
≥HS Diploma ≤ Bachelor’s Degree ^‡^	347	21.1	198 (18.8)	19 (20.9)	18 (29.0)	112 (25.8)	
≥Bachelor’s Degree ^§^	1061	64.7	729 (69.2)	51 (56.0)	32 (51.6)	249 (57.4)	
Race (*N* = 1648)							
American Indian or Alaskan Native	24	1.5 *	13 (1.2)	3 (3.3)	1 (1.6)	7 (1.6)	2.59 (3), *p* = 0.459
Asian	130	7.9 *	91 (8.6)	4 (4.4)	7 (11.1)	28 (6.4))	4.55 (3), *p* = 0.208
Black or African American	83	5.0 *	29 (2.7)	13 (14.3)	5 (7.9)	36 (8.2)	38.24 (3), *p* < 0.001
Native Hawaiian or Other Pacific Islander	6	0.4 *	6 (.6)	0 (0.0)	0 (0.0)	0 (0.0)	3.38 (3), *p* = 0.337
White	1349	81.9 *	896 (84.4)	67 (73.6)	44 (69.8)	342 (78.1)	20.84 (3), *p* < 0.001
Other	112	6.8 *	62 (5.9)	7 (7.7)	9 (14.3)	34 (7.8)	7.77 (3), *p* = 0.051
Annual Household Income (*N* = 1464)							29.19 (18), *p* = 0.046
<$10,000	159	10.9	90 (9.5)	7 (9.0)	9 (20.0)	53 (13.6)	
$10,000–$29,999	461	31.5	287 (30.1)	24 (30.8)	17 (37.8)	133 (34.2)	
$30,000–$39,999	216	14.8	135 (14.2)	14 (17.9)	7 (15.6)	60 (15.4)	
$40,000–$49,999	154	10.5	100 (10.5)	8 (10.3)	4 (8.9)	42 (10.8)	
$50,000–$75,999	242	16.5	163 (17.1)	12 (15.4)	7 (15.6)	60 (15.4)	
$76,000–$99,999	104	7.1	78 (8.2)	5 (6.4)	1 (2.2)	20 (5.1)	
$100,000 or more	128	8.7	99 (10.4)	8 (10.3)	0 (0.0)	21 (5.4)	
Sex (*N* = 1648)							31.83 (3), *p* < 0.001
Female	891	54.1	517 (49.0)	59 (64.8)	43 (68.3)	272 (62.1)	
Male	757	45.9	539 (51.0)	32 (35.2)	20 (31.7)	166 (37.9)	
Ethnicity (*N* = 1648)							11.99 (3), *p* = 0.007
Not of Hispanic, Latino, or Spanish origin	1324	80.3	873 (82.7)	71 (78.0)	44 (69.8)	336 (76.7)	
Hispanic, Latino, or Spanish origin	324	19.7 *	183 (17.3)	20 (22.0)	19 (30.2)	102 (23.2)	
Mexican, Mexican American, Chicano	245	14.9 *	137 (13.0)	14 (15.4)	16 (25.4)	78 (17.8)	11.53 (3), *p* = 0.009
Puerto Rican	11	0.7 *	5 (0.5)	3 (3.3)	0 (0.0)	3 (0.7)	10.51 (3), *p* = 0.015
Cuban	14	0.8 *	9 (0.9)	1 (1.1)	0 (0.0)	4 (0.9)	0.628 (3), *p* = 0.890
Other Hispanic, Latino, or Spanish origin	87	5.3 *	50 (4.7)	3 (3.3)	5 (7.9)	29 (6.6)	3.81 (3), *p* = 0.283

* Respondents were able to represent their racial and ethnic heritage by selecting more than one racial or ethnic group. For this table, the percentages reflect a denominator of 1648 for each of these response options and may not total 100%. ^†^ Includes respondents who self-identified as attaining an education level of “Less than some high school”, “Some high school”, and “High School Diploma, GED, or equivalent”; ^‡^ Includes respondents who self-identified as attaining an education level of “Trade school”, “Some college”, and “Associate degree”; ^§^ Includes respondents who self-identified as attaining an education level of “Bachelor’s Degree” or “Graduate Degree”.

**Table 3 ijerph-19-00368-t003:** Unadjusted and adjusted prevalence odds ratios for “Plan to get the COVID-19 vaccine when it is available” among study respondents.

	Plans to Get the COVID-19 Vaccine When Available									
	Yes		No, Not Sure, Prefer Not to Say		Unadjusted Prevalence Odds Ratio Yes vs. No, Not Sure, Prefer Not to Say			Adjusted Prevalence Odds Ratio Yes vs. No, Not Sure, Prefer Not to Say		
Factor	*n*	%	*n*	%	POR	95% CI	*p*-Value	POR	95% CI	*p*-Value
Sex	1056	100.0	592	100.0						
Male	539	51.0	218	36.8	1.0			1.0		
Female	517	49.0	374	63.2	0.559	0.455, 0.687	<0.001	0.650	0.518, 0.816	<0.001
Age (years)	1056	100	592	100						
50+	498	47.2	240	40.5	1.0			1.0		
35–49	332	31.4	227	38.3	0.705	0.561, 0.886	0.003	0.689	0.534, 0.890	0.004
18–34	226	21.4	125	21.1	0.871	0.667, 1.14	0.313	0.867	0.641, 1.171	0.352
Hispanic, Latino, or Spanish Ethnicity	1056	100.0	1324	100.0						
No	873	82.7	451	76.2	1.0			1.0		
Yes	183	17.3	141	23.8	0.670	0.524, 0.859	0.001	0.826	0.622, 1.096	0.186
Black or African American	1056	100.0	592	100.0						
No	1027	97.3	538	90.9	1.0			1.0		
Yes	29	2.7	54	9.1	0.281	0.177, 0.447	<0.001	0.351	0.211, 0.584	<0.001
Education Level	1054	100.0	587	100.0						
≥4-year college	658	62.4	271	46.2	1.0			1.0		
>HS < 4-year college	269	25.5	210	35.8	0.528	0.420, 0.664	<0.001	0.572	0.442, 0.739	<0.001
≤HS	127	12.0	106	18.1	0.493	0.368, 0.662	<0.001	0.565	0.401, 0.795	0.001
Household Income	952	100.0	512	100						
$100,000 or more	99	10.4	29	5.7	1.0			1.0		
$76,000–$99,999	78	8.2	26	5.1	0.879	0.479, 1.61	0.68	0.939	0.506, 1.741	0.841
$50,000–$75,999	163	17.1	79	15.4	0.604	0.369, 0.99	0.044	0.687	0.415, 1.136	0.144
$40,000–$49,999	100	10.5	54	10.5	0.543	0.319, 0.922	0.023	0.682	0.395, 1.178	0.170
$30,000–$39,999	135	14.2	81	15.8	0.488	0.297, 0.803	0.004	0.623	0.373, 1.041	0.071
$10,000–$29,999	287	30.1	174	34.0	0.483	0.301, 0.761	0.001	0.632	0.395, 1.011	0.056
<$10,000	90	9.5	69	13.5	0.382	0.227, 0.642	<0.001	0.565	0.327, 0.976	0.041

## Data Availability

Deidentified data can be requested from the corresponding author.

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
