# Peer review of "Sociodemographic Factors Associated with Vaccine Hesitancy in Central Texas Immediately Prior to COVID-19 Vaccine Availability"

_ijerph, 2021, doi:10.3390/ijerph19010368_

Round 1

Reviewer 1 Report

The manuscript tackles interesting topic of social and demographic factors that affect the willingness to obtain the COVID-19 vaccine and vaccine hesitancy. The unquestionable strength of the study is that the topic is very important and timely and the research itself was designed and described clearly. Moreover, as it identifies the main sociodemographic determinants that affect the willingness to be vaccinated against COVID-19, it provides insights for the future vaccination programmes and has great value, especially as we face another wave of the COVID-19 pandemic. However, I believe that the paper needs some minor improvements before it could be considered for publication in the Journal:

  1. I believe that in the ‘Introduction’ apart from explaining key issues related to vaccine hesitancy the Authors could provide some more information regarding the public’s general knowledge and attitudes toward vaccination and, COVID-19 vaccine in particular.

  1. More detailed information about the discourse the pros and cons of the vaccine could be also described.

  1. The Authors could also briefly describe the vaccine strategy of the US government and/or Texas authorities: how was it planned and undertaken? Is it successful? What is the vaccine rate in Texas and does it differ from that in other states in the US?

  1. While describing the main reasons for vaccine hesitancy (lines 52-62) the Authors should also describe that for some communities and/or individuals it resulted from moral concerns related to the fact that some vaccine manufacturers used abortion-derived fetal cell lines:

-- Zimmerman RK. Helping patients with ethical concerns about COVID-19 vaccines in light of fetal cell lines used in some COVID-19 vaccines. Vaccine. 2021;39(31):4242-4244. doi:10.1016/j.vaccine.2021.06.027

-- Garcia LL, Yap JFC. The role of religiosity in COVID-19 vaccine hesitancy. J Public Health (Oxf). 2021 Sep 22;43(3):e529-e530. doi: 10.1093/pubmed/fdab192.

  1. Finally, while the Authors rightly observe that vaccine hesitancy is of particular concern among communities of color (lines 63) it should be clearly indicated that lower levels of trust toward science and state-sponsored health program among such ethnic minorities as African-Americans, Mexican-Americans, native Americans, Hawaii and Alaskan Natives, often results from their previous experiences with unethical healthcare research in ethnic populations (colonization, eugenics and medical experiments), their experiences with systemic racism and discrimination, under-representation of minorities in health research and vaccine trials or negative experiences within a culturally insensitive healthcare system.

-- Razai MS, Osama T, McKechnie DGJ, Majeed A. Covid-19 vaccine hesitancy among ethnic minority groups. BMJ. 2021 Feb 26;372:n513. doi: 10.1136/bmj.n513. 

-- Kadambari S, Vanderslott S. Lessons about COVID-19 vaccine hesitancy among minority ethnic people in the UK. Lancet Infect Dis. 2021 Sep;21(9):1204-1206. doi: 10.1016/S1473-3099(21)00404-7. 

-- Reid JA, Mabhala MA. Ethnic and minority group differences in engagement with COVID-19 vaccination programmes - at Pandemic Pace; when vaccine confidence in mass rollout meets local vaccine hesitancy. Isr J Health Policy Res. 2021 May 27;10(1):33. doi: 10.1186/s13584-021-00467-9. 

  1. I am surprised that apart from age, gender, race, ethnicity, level of educational attainment, and income no information on such important demographic data as: domicile, family status, religion/confession and self-perceived status health was given which also might have influenced respondents attitudes towards vaccination.

  1. Although in the ‘Discussion’ the Authors do mention some studies on the vaccine hesitancy this section would benefit from putting these findings in broader perspective and comparing it with the situation in other countries.

-- Machingaidze, S., Wiysonge, C.S. Understanding COVID-19 vaccine hesitancy. Nat Med 27, 1338–1339 (2021). https://doi.org/10.1038/s41591-021-01459-7

-- Bass SB, Wilson-Genderson M, Garcia DT, Akinkugbe AA, Mosavel M. SARS-CoV-2 Vaccine Hesitancy in a Sample of US Adults: Role of Perceived Satisfaction With Health, Access to Healthcare, and Attention to COVID-19 News. Front Public Health. 2021 Apr 29;9:665724. https://doi.org/10.3389/fpubh.2021.665724.

-- Hudson A, Montelpare WJ. Predictors of Vaccine Hesitancy: Implications for COVID-19 Public Health Messaging. Int J Environ Res Public Health. 2021 Jul 29;18(15):8054. https://doi.org/10.3390/ijerph18158054.

-- AlShurman BA, Khan AF, Mac C, Majeed M, Butt ZA. What Demographic, Social, and Contextual Factors Influence the Intention to Use COVID-19 Vaccines: A Scoping Review. Int J Environ Res Public Health. 2021 Sep 4;18(17):9342. https://doi.org/10.3390/ijerph18179342.

  1. Finally, I believe that the Authors could provide some recommendations suggesting possible guidelines that should be implemented in order to further overcome vaccine hesitancy.

All in all, while I believe that the research topic important and timely the manuscript requires some minor modifications and another review.

Reviewer 2 Report

Nowadays, there is a need for such papers as this one. Irrespectively of country, vaccination is our only way of getting rid of the pandemic and hence there is merit in this work. The topics, methods and results are clearly presented, the paper focuses on the statistical model and its suggestions, forming them into conclusions. There is mainly one part, where the paper should be improved.

As the authors rightfully note, the sample is not representative, but they should contribute a section on describing the degree of non-representativeness, e.g. information on the ethnic distribution, age and income distribution of this area. 

Besides this comment, there are no other points that have to be improved.

Reviewer 3 Report

I have some minor comments to improve the paper and its utility when published. These include:a

a) Methodology:

i) How was the questionnaire developed - there are currently no details (unless I missed this), e.g. based on the literature, etc., in this area (current and previous pandemics)? Was it piloted before full use to enhance its robustness/ validity, etc?

ii) How were the respondents selected (if appropriate)? 

b) Results - what % does 1,660 persons represent of those approached/ contacted to take part in this survey as response rates are generally low to surveys such as this?

c) Discussion 

i) It is now nearly a year since the survey was undertaken and we now have full roll-outs of the vaccination programme - are you see the findings in practice among those citizens coming forward to be vaccinated (it would be very useful to know the difference between what citizens said they would do and the reality in practice to prevent deaths among themselves/ family members, etc.)?

ii) Building on (i) - what do you now recommend for the Health Plan to improve vaccination rates where concerns - especially as now nearly a year since the end of the survey 

Reviewer 4 Report

In this paper, Litaker et al describe an online survey performed by a health insurance company in Central Texas, to investigate intention to receive COVID-19 vaccine among its policy holders, prior to the release of two mRNA vaccines in the USA. A total of 1648 responses were analysed, and overall, the authors report that Black or African American race, female sex, age 35 to 49 years, low income (<10,000 USD) and low level of education were all associated with greater vaccine hesitancy.

Overall, the results are not surprising, but confirming these socio-demographic risk factors for vaccine hesitancy is, in itself, a useful finding, as it may help to guide specific interventions to counter this phenomenon and move these vaccine hesitant populations away from the “middle ground” and towards the “vaccine acceptance” end of the spectrum.

Overall, the authors are to be commended for the quality of the survey and the report, which is succinct and informative. A major limitation, which is adequately acknowledged by the authors, is that the study population are all individuals in a single health insurance organisation, (and people who actually have insurance!), and in this regard, they may differ from the general population. However, there is not much that the authors can do to change this, and the results remain of interest nonetheless, of course taken in the context to which they apply.

I have no major comments on this paper.

Author Response

Thank you for your feedback and comments.